# CB2 Receptor Stimulation and Dexamethasone Restore the Anti-Inflammatory and Immune-Regulatory Properties of Mesenchymal Stromal Cells of Children with Immune Thrombocytopenia

**DOI:** 10.3390/ijms20051049

**Published:** 2019-02-28

**Authors:** Francesca Rossi, Chiara Tortora, Giuseppe Palumbo, Francesca Punzo, Maura Argenziano, Maddalena Casale, Alessandra Di Paola, Franco Locatelli, Silverio Perrotta

**Affiliations:** 1Department of Women, Child, and General and Specialized Surgery, University of Campania Luigi Vanvitelli, 80138 Naples, Italy; chiara.tortora@unicampania.it (C.T.); francesca.punzo19@gmail.com (F.P.); maurargenziano@gmail.com (M.A.); maddalena.casale@unicampania.it (M.C.); alessandra.dipaola92@gmail.com (A.D.P.); silverio.perrotta@unicampania.it (S.P.); 2Department of Haematology, Bambino Gesù Hospital, 00165 Rome, Italy; giuseppe.palumbo@opbg.net (G.P.); franco.locatelli@opbg.net (F.L.); 3Department of Experimental Medicine, University of Campania Luigi Vanvitelli, 80138 Naples, Italy

**Keywords:** immune thrombocytopenia, mesenchymal stromal cells, cannabinoid receptor 2, dexamethasone

## Abstract

Immune thrombocytopenia (ITP) is an autoimmune disorder characterized by antibody-mediated platelet destruction, with a complex and unclear pathogenesis. The impaired immunosuppressive capacity of mesenchymal stromal cells in ITP patients (ITP-MSCs) might play a role in the development of the disease. Correcting the MSC defects could represent an alternative therapeutic approach for ITP. High-dose dexamethasone (HD-Dexa) is the mainstay of the ITP therapeutic regimen, although it has several side effects. We previously demonstrated a role for cannabinoid receptor 2 (CB_2_) as a mediator of anti-inflammatory and immunoregulatory properties of human MSCs. We analyzed the effects of CB_2_ stimulation, with the selective agonist JWH-133, and of Dexa alone and in combination on ITP-MSC survival and immunosuppressive capacity. We provided new insights into the pathogenesis of ITP, suggesting CB_2_ receptor involvement in the impairment of ITP-MSC function and confirming MSCs as responsive cellular targets of Dexa. Moreover, we demonstrated that CB_2_ stimulation and Dexa attenuate apoptosis, via Bcl2 signaling, and restore the immune-modulatory properties of MSCs derived from ITP patients. These data suggest the possibility of using Dexa in combination with JWH-133 in ITP, reducing its dose and side effects but maintaining its therapeutic benefits.

## 1. Introduction

Immune thrombocytopenia (ITP) is a multifactorial autoimmune disorder characterized by antibody-mediated platelet destruction [1,2,3]. Due to the complexity of the immune system and the heterogeneity of the disease, the pathogenesis of ITP remains unclear [4]. However, ITP is characterized by several aberrations such as the production of platelet autoantibodies, polarization of macrophages towards the M1 phenotype, an elevated Th1/Th2 ratio, and alteration of circulating cytokine profile [5,6,7]. Recent evidences suggest that mesenchymal stromal cells (MSCs) could also be involved in ITP pathogenesis [8]. MSCs are multipotent cells that play an important role in immune modulatory functions. They can inhibit both T- and B-lymphocyte proliferation and activation. Inhibitory function of MSCs is dependent on cell–cell contact and on the release of soluble factors that inhibit lymphocyte response [9,10].

Several studies have shown that in MSCs from ITP patients (ITP-MSCs) the immunoregulative abilities are compromised. In particular, ITP-MSCs show enhanced senescence and apoptosis with a reduced capacity to inhibit T-cell proliferation [11,12,13].

In literature, it is widely reported that high-dose dexamethasone (HD-Dexa) may be the first line for management of ITP [14,15], although it has several side effects [16,17]. It is known that HD-Dexa induces an increase of M2-like macrophages that suppress autoimmunity, and is able to elevate the Th1/Th2 ratio [18,19].

Previous studies described a restoration of cytokine balance after HD-Dexa treatment and an influence of glucocorticoids on the immunosuppressive activity of MSCs [20,21]. In 2013, we demonstrated a role for cannabinoid receptor 2 (CB_2_) as a mediator of anti-inflammatory and immunoregulatory properties of human MSCs [22].

The CB_2_ receptor is involved in immune regulation by suppressing immune cell activation through the modulation of T helper cells [23], the inhibition of pro-inflammatory cytokine production [24], and nuclear factor-kappa B (NF-κB)-dependent apoptosis [25]. Moreover, we associated a CB_2_ functional variant with several inflammatory/immune-based diseases such as childhood ITP [26].

Based on these evidences, in the present study, we evaluated whether JWH-133, a selective agonist at CB_2_ receptor, and Dexa are able to restore the immunomodulatory properties of MSCs in ITP patients in order to suggest new therapeutic approaches.

## 2. Results

### 2.1. Expression of CB2 Receptor in MSCs Derived from ITP Patients

We performed a real-time PCR and a Western blot (WB) to evaluate the mRNA expression levels and protein density of the CB_2_ receptor in MSCs isolated from the bone marrow (BM) of ITP patients, at different stages of MSCs growth, from Passage 2 to Passage 8 (P2 to P8) (Figure 1, Table 1A,B). We observed the highest expression of CB_2_ mRNA and protein density from P6, similarly to what we have previously described in MSCs isolated from the BM of healthy subjects (MSC-CTR) [22]. Therefore, we compared the CB_2_ expression of ITP-MSCs with CTR-MSCs at P6 (Figure 2, Table 2A,B). Molecular and biochemical analysis revealed that, in ITP-MSCs, CB_2_ was significantly lower than CTR-MSCs, indicating that a reduction of CB_2_ expression could be related to ITP-MSCs’ impaired function.

### 2.2. Effects of JWH-133 and Dexa on Cytokine Release

The multi-ELISA assay (Table 3) revealed a significant increase of the pro-inflammatory cytokine interleukin 6 (IL-6) in supernatants of ITP-MSCs with respect to MSCs-CTR. In parallel, ITP-MSCs showed a significant decrease of the anti-inflammatory cytokine, interleukin 4 (IL-4).

Both JWH-133 (2.5 µM) and Dexa (100 nM), alone and in combination, are able to restore IL-6 balance—the former in a less marked manner. The co-administration of JWH-133 (2.5 µM) with Dexa (100 nM) also induces a significant reduction of the pro-inflammatory cytokine, similar to the one induced by Dexa (100 nM) administration.

JWH-133 (2.5 µM) and Dexa (100 nM) are able to induce an increase of IL-4 that is greater when used in combination and comparable to the increases of CTR-MSCs.

Moreover, to confirm the CB_2_ anti-inflammatory properties in ITP-MSCs, we blocked CB_2_ with the reverse agonist AM630 (1 µM) that respectively increased and decreased IL-6 and IL-4 release, and significantly counteracted the effects induced by JWH-133.

### 2.3. Effects of JWH-133 and Dexa on ITP-MSCs Apoptosis

To evaluate whether JWH-133 (2.5 µM) and Dexa (100 nM) could protect ITP-MSCs against apoptosis, we performed a cytofluorimetric assay (Table 4A) and a WB (Table 4B and Figure 3) to underline any differences in Bcl2 protein density. JWH-133 (2.5 µM) and Dexa (100 nM), alone and in combination, significantly reduced the percentage of total apoptotic cells. Accordingly, the WB revealed that Bcl2 protein density increased after treatments and co-treatments.

Moreover, to demonstrate whether CB_2_ is really required for survival of ITP-MSCs, we performed an apoptotic assay also after CB_2_ blockage with AM630 (1 µM). As expected, the inverse agonist induced an increase of the total apoptotic cells percentage with respect to the untreated cells and counteracted the positive effects on cellular survival induced by JWH-133 (2.5 µM) in ITP-MSCs (Appendix A).

### 2.4. Effects of JWH-133 and Dexa on ITP-MSCs’ Immunosuppression Capacity

To evaluate whether JWH-133 (2.5 µM) and Dexa (100 nM), restored the impaired immunosuppression capacity in ITP-MSCs, we co-cultured MSCs from ITP patients with T-cells obtained by the stimulation of peripheral blood mononuclear cells (PBMCs), isolated from healthy subjects, with PHA.

As shown in Table 5, ITP-MSCs were unable to prevent T-cell proliferation. The capacity of ITP-MSCs in reducing T-lymphocyte number was significantly higher after JWH-133 (2.5 µM) and Dexa (100 nM) administration. The same effect was observed when the two drugs were used in combination. The CB_2_ blockage with AM630 (1 µM) significantly counteracted the JWH-133 (2.5 µM) effects confirming the important role of the receptor as a mediator of MSCs’ immunosupporessive properties (Appendix A). We also evaluated the levels of the pro-inflammatory cytokine, tumor necrosis factor alpha (TNF-α) (Table 6), in supernatants of T-cells, alone and co-cultured with ITP-MSCs, stimulated or not with lipopolysaccharide (LPS) (500 ng/mL). We did not observe differences in TNF-α levels between T-cell ITP-MSCs co-culture compared to T-cells alone, indicating that the MSCs from ITP patients had lost their capacity to inhibit TNF-α release by T-cells.

The same trend was observed in T-cells-ITP-MSCs co-culture treated with LPS (500 ng/mL) respect to untreated co-culture, although as expected, the levels of TNF-α were higher after the inflammatory stimulus with LPS (500 ng/mL). Both JWH-133 (2.5 µM) and Dexa (100 nM) were able to restore the inhibitory properties of ITP-MSCs, reducing TNF-α release and showing a strong synergic effect when used in combination. Moreover, to understand which of the two cell types released TNF-α, we compared their release in a basal condition and after LPS stimulation, and, as shown in Appendix A, MSCs released a negligible amount of TNF-α, with respect to T-cells, after inflammatory stimulus.

## 3. Discussion

Immune thrombocytopenia (ITP) is a multifactorial autoimmune disorder characterized by antibody-mediated platelet destruction [1,2,3]. Impairments of T-cells and macrophages, together with a cytokine imbalance, have been recognized as key processes, although the pathogenesis remains unclear [4,5,6,7]. However, there is an increasing interest in MSC involvement in ITP pathogenesis [11] due to its important role in immune modulatory functions [8,9,10].

In MSCs from ITP patients, proliferation is impaired, and immunosuppressive capacity is compromised [12,13]. Because their impairment might play a role in the development of ITP, correcting the defects of MSCs could represent a potential alternative therapeutic approach for ITP. Glucocorticoid (GC) therapy and high-dose dexamethasone (HD-Dexa) are the mainstays of an ITP therapeutic regimen [14,15].

Dexa, a synthetic GC, has a strong anti-inflammatory effect and modulates several aspects of ITP pathogenesis [18,19,20], but its application is limited by important side effects [16,17]. Therefore, a novel therapeutic approach that combines the classical immunosuppressive therapy with new pharmacological treatments would be desirable. In 2013, we demonstrated a role for the cannabinoid receptor 2 (CB_2_) as a mediator of anti-inflammatory and immunoregulatory properties of human MSCs [22]. The CB_2_ receptor is involved in immune regulation by suppressing immune cell activation through the modulation of T helper cells [23], the inhibition of pro-inflammatory cytokine production [24], and Nuclear factor-kappa B (NF-kB)-dependent apoptosis [25]. Moreover, we associated a CB_2_ functional variant with several inflammatory/immune-based diseases such as childhood ITP [26]. In the present study, we evaluated the effects of CB_2_ stimulation, with a selective agonist (JWH-133), and those of Dexa on ITP-MSC survival and immunosuppressive capacity.

In our previous study, we observed that in healthy MSCs [22], in addition to ITP-MSCs, CB_2_ reaches a higher expression at Passage 6 (P6).

Interestingly, in ITP-MSCs, CB_2_ receptor was significantly lower than in CTR-MSCs at P6, suggesting that its reduction could be related to ITP-MSCs’ impaired function. This result is in agreement with the well-known role of the cannabinoid receptor in immune regulation [25] and provides new insight into the pathogenesis of ITP.

Considering that circulating cytokines reflect the status of immune processes [27], the evaluation of their profile could be useful to investigate the effect of a pharmacological therapy. In our study, ITP-MSCs showed an abnormal cytokine secretion, which may contribute to their reduced function. In particular, we observed an increased production of the pro-inflammatory cytokine IL-6.

The selective CB_2_ stimulation exerted anti-inflammatory activity, restoring the same physiological cytokine balance observed with Dexa. Interestingly, we also observed a similar anti-inflammatory effect when the two drugs were used in combination, confirming our hypothesis on the possible beneficial effect induced by the co-administration of GC with a complementary agent.

MSCs from ITP patients show a slow proliferative capacity and a high rate of apoptosis [12]. Accordingly, we observed that ITP-MSCs in vitro expanded slower, appeared larger and more flattened than CTR-MSCs, and showed a high percentage of cells in total apoptosis. To further investigate the underling mechanisms involved in these changes, we analyzed the expression of Bcl2, a key protein in the intrinsic pathway of apoptosis [28], in ITP-MSCs treated with JWH-133 and Dexa.

We observed an overexpression of Bcl2 that could explain the decrease of the apoptosis rate observed in ITP-MSCs after treatments and co-treatment.

T-cells are considered the primary targets of the immunosuppressive activity of MSCs [10]. In physiological condition, MSCs can inhibit lymphocyte proliferation and activation [29]. T-cells from ITP patients show a significant clonal expansion and a defective secretion of TNF-α [5,9,30]. The pro-inflammatory cytokine, TNF-α, produced primarily by immune cells such as T lymphocytes, is the major participant in the initiation and orchestration of complex events in inflammation and immunity [31].

Moreover, children with ITP have elevated levels of TNF-α [32]. For these reasons, we co-cultured MSCs from ITP patients with healthy PHA activated T-cells and analyzed lymphocytes viability and tumor necrosis factor-alpha (TNF-α) release after treatments. Dexa was able to restore the inhibitory properties of ITP-MSCs on T-cells by reducing their viability. These data are in accordance with the well-identified primary role of GC in suppressing excessive T-cell [19] growth and in particular with the study of Buron et al., in which MSCs treated with Dexa are able to inhibit lymphocyte proliferation [33]. Interestingly, with CB_2_ receptor stimulation we observed a more marked reduction of T-cell viability. Considering that MSCs released a negligible amount of TNF-α respect to T-cells, also after inflammatory stimulus, we hypothesized that the quantified TNF-α in co-culture was attributable to T-cells. Obviously, further investigations are needed to support this hypothesis.

Our data showed that ITP-MSCs were defective in inhibiting TNF-α secretion by T-cells, and the co-administration of Dexa and JWH-133 reversed this effect in a more evident manner than the single administration. This result, together with their common ability to modulate NF-κB, a transcription factor for pro-inflammatory mediators [25,34], suggests a hypothetical synergism between Dexa and JWH-133. Nevertheless, considering that both drugs are known to modulate T-cell activity [19,23,35], this observation, instead of a real synergism, could rather be the result of a direct effect on T-cells and of an MSC-mediated effect. Certainly further investigations are needed to clarify this possible, as yet confirmed, interaction.

In conclusion, we provided new insight into the pathogenesis of ITP, suggesting CB_2_ receptor involvement in the impairment of ITP-MSC function. For the first time, it is shown that CB_2_ stimulation acts as a mediator of MSCs’ immunosupporessive properties in a compromised immune system condition. These data are strengthened by our previous study in healthy LPS stimulated MSCs [22], which mimic the inflammatory status seen in ITP-MSCs, overcoming the limited sample size. In addition, confirming the receptor as an important anti-inflammatory and immunomodulatory target, the CB_2_ blockade with the inverse agonist, AM630, counteracted the effects induced by its selective stimulation.

Moreover, we also confirm MSCs as responsive cellular targets of Dexa [36], considering that we observed an improvement of ITP-MSC function after treatment with this drug.

In particular, we demonstrated that Dexa and CB_2_ stimulation attenuated apoptosis via Bcl-2 signaling, and restored the immune-modulatory properties of MSCs derived from ITP patients.

These data suggest the possibility of using Dexa in combination with CB_2_ stimulation in ITP, reducing its dose and side effects and maintaining its therapeutic benefits. In the future, other in vitro and in vivo studies with the aim of translating our findings into clinical practice are required.

## 4. Materials and Methods

### 4.1. Source of MSCs

This study was performed using MSCs isolated from the BM of two newly diagnosed ITP children (median age 7 ± 2 years) and two healthy donors (median age 7 ± 2 years). ITP patients were enrolled in the Department of Women, Child and General and Specialized Surgery of University of Campania Luigi Vanvitelli and were free of any glucocorticosteroids therapy for at least 3 months. Healthy donors were enrolled in Hematology Department of Bambino Gesù Hospital. All procedures performed in this study were in accordance with the Helsinki Declaration of Principles, the Italian National Legislation, and the Ethics Committee of the University of Campania Luigi Vanvitelli, which formally approved the study (Identification code 499, 12 September 2017). Written informed consent was obtained from parents, and assent was acquired from children before any procedures.

### 4.2. Isolation, Expansion, and Characterization of MSCs

Mononuclear cells were isolated from BM by density gradient centrifugation (Ficoll 1.077 g/mL; Lympholyte, Cedarlane Laboratories Ltd., Uden, The Netherlands) and plated in non-coated T25 polystyrene culture flasks (Corning Costar, Celbio, Milan, Italy). The complete culture medium consisted of a low-glucose Dulbecco’s Modified Eagle Medium (LG-DMEM) (Euroclone, Milan, Italy) supplemented with 10% fetal bovine serum (FBS; Gibco, Life Technologies Ltd., Paisley, UK), 50 U/mL penicillin, 50 mg/mL streptomycin, and 2 mM l-glutamine (Euroclone, Pero, MI, Italy). Non-adherent cells were discarded after 48 h, while adherent cells were maintained at 37 °C in a humidified atmosphere containing 5% CO_2_. The culture medium was replaced twice a week. After reaching 80% confluence, MSCs were split, re-plated (4000 cells/cm^2^) for expansion, and harvested until the eighth passage (P0–P8). MSC mRNA and proteins were isolated from each passage to analyze CB2 expression and density. MSCs were characterized by flow cytometry using monoclonal antibodies, conjugated with fluorescein isothiocyanate (FITC) or phycoerythrin (PE), specific for the following antigens: HLA DR, HLA A-B-C, CD45, CD34, CD13, CD14, CD31, CD80, CD90 (BD PharMingen, San Diego, CA, USA), CD73, and CD105 (Serotec, Kidlington, Oxford, UK). Labeled cells were acquired using a FACSCanto flow cytometer (BD PharMingen), and data were elaborated using FACSDiva software (TreeStar Inc., Ashland, OR, USA).

### 4.3. MSCs and T Cells Co-Culture

PBMCs were isolated using Ficoll density gradient centrifugation from venous blood obtained from two healthy subjects (median age 7 ± 2 years) and were cultured in Roswell Park Memorial Institute (RPMI) 1640 medium supplemented with 10% FBS, 2 mM l-glutamine, and 100 U/mL penicillin/streptomycin with or without mitogen-induced stimulation by phytohemagglutinin (PHA). Activated lymphocytes (1.25 × 10^5^ cells/well) were then co-cultured with MSCs (2.5 × 10^4^ cells/well) on a 24-well plate at a ratio of 5:1 in the RPMI medium and treated with LPS (500 ng/mL) to mimic inflammatory condition. After 3 days, JWH-133 (2.5 µM) and Dexa (100 nM), alone or in combination, were added to the plates for 24 h. The supernatants of co-cultures were collected to analyze TNF-α release with an enzyme-linked immunosorbent assay (ELISA), while T-cells were used to perform the cell viability assay.

### 4.4. Total RNA Extraction and Reverse Transcription Quantitative Polymerase Chain Reaction (RTqPCR)

Total RNA from MSC cultures was extracted using Qiazol® (Qiagen, Hilden, Germany) following the manufacturer’s instructions. EasyScript™ cDNA Synthesis Kit (abm, Foster City, California, USA) was used to synthesize from approximately 1000 ng of mRNA, the first-strand cDNA. The transcript levels of CB2 were detected by RT-qPCR using a CFX96 Real-Time PCR system (Bio-Rad, Hercules, CA, USA) using I-Taq Universal SYBR® Green Master Mix (Bio-Rad). The cycling conditions were 10 min at 95 °C (initial denaturation) followed by 40 cycles of 15 s at 94 °C (denaturation) and 1 min at 68 °C (annealing/extension/data collection). The β-actin gene served as the reference gene for the normalization of the real-time PCR products. The PCR primers used to detect each gene were designed using the Primer 3 program and synthesized by Sigma Aldrich (CB2_F 5′-AAGGCTGTCTTCCTGCTGAA-3′, CB2_R 5′- CACAGAGGCTGTGAAGGTCA-3, β-Actin_F 5′-GCGAGAAGATGACCCAGATC-3′, β-Actin_R 5′-GGA TAGCACAGCCTGGATAG-3′). We performed the assays in technical duplicate for each subject and tested the linearity and efficiency of the experiments over dilutions of cDNA including five orders of greatness. To confirm the specificity of the reactions, we performed the dissociation curve analysis of amplification products. To analyze the data and achieve the relative gene expression levels we used the 2^−∆∆*C*t^ method.

### 4.5. Protein Isolation; Western Blot

Proteins were extracted from MSC cultures using a RIPA lysis buffer (Millipore) and following the manufacturer’s instructions. CB2 and Bcl2 proteins were characterized in total lysates from cell cultures by Western blotting. Membranes were incubated overnight at 4 °C with a rabbit polyclonal anti CB2 antibody (1:500 dilution; Abcam catalogue number 3561 tested in CB_2_ KO mice [37]) and mouse monoclonal anti Bcl-2 (1:200 dilution; Santa Cruz). Reactive bands were detected by chemiluminescence (Immobilon Western Millipore) on a C-DiGit® Blot scanner (LI-COR Biosciences). A mouse polyclonal anti β-Tubulin antibody (1:5000 dilution; Elabscience) was used to check for comparable protein loading and as a housekeeping protein. We performed single experiments on each one of the two ITP patients’ samples (displayed as mean ± SD). Images were captured, stored, and analyzed using Image Studio Digits software, version 5.0.

### 4.6. ELISA

Supernatants were obtained from MSC culture as well as MSC–T-cell co-cultures.

IL-6, TNF-α, and IL-4 levels were measured using a commercially available Human Inflammatory Cytokines Multi-Analyte ELISArray Kit (Qiagen, QIAGEN S.p.A., Italy) according to the manufacturer’s instructions. Briefly, a microplate was coated with monoclonal antibodies that were specific to the cytokines. Standards and supernatants were pipetted into the wells of the microplate. A positive control was obtained by pipetting only the standard into the wells. A negative control was obtained by pipetting the standard and cell culture supernatants into non-coated wells. After the plate was washed, enzyme-linked polyclonal antibodies specific for IL-6, TNF-α, and IL-4 were added to the wells. The reaction was revealed by the addition of the substrate solution. The optical density was measured at a wavelength of 450 nm by using the Tecan Infinite M200 (Tecan, Switzerland) spectrophotometer. Cytokines concentrations (pg/mL) were determined against a standard concentration curve. All samples were run in duplicate.

### 4.7. Annexin V & Dead Cell Assay Kit

Apoptosis in treated MSCs was evaluated by a fluorometric assay with the Muse cell analyzer machine with the Annexin V & Dead Cell Assay Kit. Test was performed after 24 h of compound exposure. Active compounds were added alone or in combination at the following concentrations: JWH-133 (2.5 µM) and Dexa (100 nM). The Muse Annexin V & Dead Cell Assay utilizes Annexin V to detect phosphatidylserine (PS) on the external membrane of apoptotic cells. A dead cell marker, 7-amino-actinomycin D (7-AAD), is also used as an indicator of cell membrane structural integrity. Briefly, 100 μL of a cell suspension (1 × 10^5^ cells/mL) was mixed with 100 μL of the Muse Annexin V & Dead Cell Reagent and incubated for 20 min at room temperature in the dark. The results, automatically displayed, were analyzed with Muse 1.4 analysis software for data acquisition and analysis.

### 4.8. Count and Viability Assay Kit

To perform the count and viability assay, we isolated the T-cells from the co-culture media. MSCs grow in adhesion, so they remained in the plate. T-cell count and viability were evaluated after 24 h of compound exposure with the Muse cell analyzer machine with Count & Viability Assay Kit. The Muse Count & Viability reagent differentially stains viable and non-viable cells based on their permeability to the two DNA binding dyes present in the reagent. Fifty microliters of a T-cell suspension (1 × 10^5^ cells/mL) was mixed with 450 μL of the Muse Count & Viability reagent and incubated for 5 min at room temperature. The results, automatically displayed, were analyzed with Muse 1.4 analysis software for data acquisition and analysis.

### 4.9. Drugs and Treatments

LPS (Sigma Aldrich), JWH-133, Dexa, and AM630 (Tocris, Avonmouth, UK) were dissolved in PBS containing DMSO. DMSO final concentration on cultures was 0.01%. CTR-MSCs, ITP-MSCs, and ITP-MSC–T-cell co-cultures were treated with LPS (500 ng/mL), JWH-133 (2.5 µM), Dexa (100 nM), and AM630 (1 µM) alone or in combination for 24 h. AM630 was applied for 15 min before JWH-133 treatment. Non-treated cultured cells were maintained in incubation media during the relative treatment time with or without vehicle (DMSO 0.01%).

### 4.10. Statistics

All the experiments were run in duplicate. Statistical analyses were performed using the non-parametric Wilcoxon test to evaluate differences between quantitative variables. A *p*-value less than 0.05 was considered statistically significant.

## Figures and Tables

**Figure 1 ijms-20-01049-f001:**
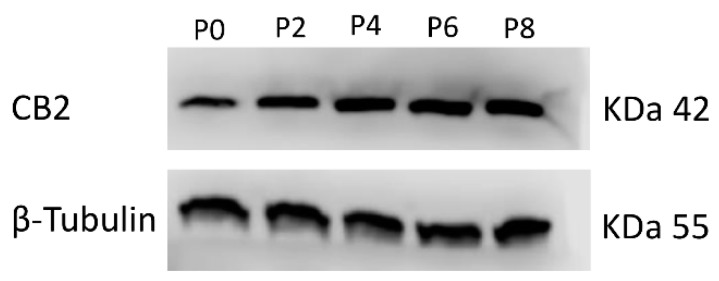
Protein density was determined by Western blotting, starting from 15 μg of total lysates. The most representative image is displayed. The proteins were detected using Image Studio Digits software.

**Figure 2 ijms-20-01049-f002:**
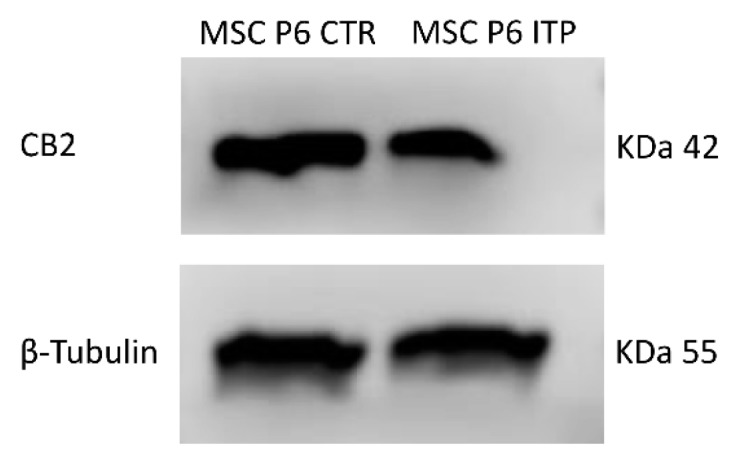
Protein density was determined by Western blotting, starting from 15 μg of total lysates. The most representative image is displayed. The proteins were detected using Image Studio Digits software.

**Figure 3 ijms-20-01049-f003:**
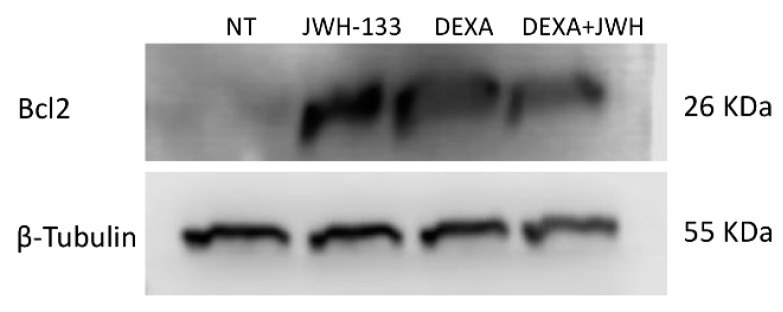
Bcl2 protein density in MSCs from two ITP patients was determined by Western blotting, starting from 15 μg of total lysates and after 24 h exposure to JWH-133 (2.5 µM) and Dexa (100 nM) alone and in combination. The most representative image is displayed. The proteins were detected using Image Studio Digits software. The intensity ratios of immunoblots compared to NT, taken as 1 (arbitrary unit), were quantified after normalizing with respective loading controls for the housekeeping protein β-tubulin and are shown in Table 4B.

**Table 1 ijms-20-01049-t001:** CB_2_ receptor expression levels in MSCs from ITP patients.

(A)	**CB2 Relative Quantification (2^−∆∆*C*t^)**				
	**Passages**				
Samples	P0	P2	P4	P6	P8
MSC ITP-1	1	2.3 *	2.5 *	4.8 *	6.0 *
MSC ITP-2	1	3.0 *	1.9 *	4.6 *	5.4 *
(B)	**CB2 Protein signal density**				
	**Passages**				
Samples	P0	P2	P4	P6	P8
MSC ITP-1	1	1.72 *	2.10 *	2.26 *	2.90 *
MSC ITP-2	1	1.80 *	2.11 *	2.29 *	2.44 *

CB_2_ mRNA expression (Table 1A) and protein density (Table 1B) in MSCs from two ITP patients at the P0, P2, P4, P6, and P8 passages. mRNA levels were determined by qPCR. Results were normalized for the housekeeping gene β-actin and are shown in Table 1A. The intensity ratios of immunoblots compared to the P0 passage, taken as 1 (arbitrary unit), were quantified after normalizing with respective loading controls for the housekeeping protein β-tubulin and are shown in Table 1B. A Wilcoxon test was used to evaluate the statistical differences in mRNA expression and protein density. * *p* ≤ 0.05 compared to the P0 passage.

**Table 2 ijms-20-01049-t002:** CB_2_ receptor expression levels in MSCs from ITP patients and healthy donors.

**(A)**	
**Samples**	**CB2 Relative Quantification (2^−∆∆*C*t^)**
MSC P6 CTR-1	1
MSC P6 CTR-2	1
MSC P6 ITP-1	0.54 *
MSC P6 ITP-2	0.62 *
**(B)**	
**Samples**	**CB2 Protein Signal Density**
MSC P6 CTR-1	1
MSC P6 CTR-2	1
MSC P6 ITP-1	0.53 *
MSC P6 ITP-2	0.49 *

CB_2_ mRNA expression (Table 2A) and protein density (Table 2B) in MSCs from two ITP patients at the P6 passage compared with levels in MSCs from two healthy donors at the same passage. mRNA levels were determined by qPCR. Results were normalized for the housekeeping gene β-actin and are shown in Table 2A. The intensity ratios of immunoblots compared to MSC P6 CTR, taken as 1 (arbitrary unit), were quantified after normalizing with respective loading controls for the housekeeping protein β-tubulin and are shown in Table 2B. A Wilcoxon test was used to evaluate the statistical differences in mRNA expression and protein density. * *p* ≤ 0.05 compared to MSC P6 CTR.

**Table 3 ijms-20-01049-t003:** IL-6 (A), and IL-4 (B) from CTR-MSCs and ITP-MSCs were investigated through a multi-ELISA assay after 24 h treatment with JWH-133 (2.5 µM), Dexa (100 nM), and AM630 (1 µM) alone and in combination.

**(A) IL-6**							
**Samples**	**CTR-1**	**ITP-1** **NT**	**ITP-1 JWH-133**	**ITP-1 DEXA**	**ITP-1** **D + J**	**ITP-1 AM630**	**ITP-1** **A + J**
pg/mL	7.04	12.87 *	9.02 ^	7.70 ^	11.01 ^	17.05 ^	15.41 ^
**Samples**	**CTR-2**	**ITP-2** **NT**	**ITP-2 JWH-133**	**ITP-2 DEXA**	**ITP-2** **D + J**	**ITP-2 AM630**	**ITP-2** **A + J**
pg/mL	7.18	16.02 *	9.83 ^	8.63 ^	9.44 ^	17.38 ^	17.52 ^
**(B) IL-4**							
**Samples**	**CTR-1**	**ITP-1** **NT**	**ITP-1 JWH-133**	**ITP-1 DEXA**	**ITP-1** **D + J**	**ITP-1 AM630**	**ITP-1** **A + J**
pg/mL	44.61	31.53 *	43.11 ^	45.26 ^	39.38 ^	28.89 ^	27.48 ^
**Samples**	**CTR-2**	**ITP-2** **NT**	**ITP-2 JWH-133**	**ITP-2 DEXA**	**ITP-2 D + J**	**ITP-2 AM630**	**ITP-2 A + J**
pg/mL	45,82	32.68 *	42.91 ^	45.03 ^	40.67 ^	29.41 ^	30.05 ^

The tables show the concentrations (pg/mL) of the two cytokines. A Wilcoxon test was used for statistical analysis. *p* < 0.05 was considered statistically significant. * vs. CTR; ^ vs. ITP NT.

**Table 4 ijms-20-01049-t004:** Apoptosis in ITP-MSCs. (Table 4A) Annexin V and PI double-stained apoptosis assay, in MSCs derived from two ITP patients after 24 h treatments with JWH-133 (2.5 µM) and dexamethasone (100 nM), alone or in combination.

**(A)**	**Percentage of Total Apoptotic ITP-MSCs**			
	**Treatments**			
**Samples**	**NT**	**JWH-133**	**DEXA**	**J + D**
MSC ITP-1	41.56	34.24 *	30.53 *	27.56 *
MSC ITP-2	44.94	31.26 *	35.27 *	31.08 *
**(B)**	**Bcl2 Protein Signal Density**			
	**Treatments**			
**Samples**	**NT**	**JWH-133**	**DEXA**	**J + D**
MSC ITP-1	1	7.94 *	10.25 *	7.85 *
MSC ITP-2	1	8.48 *	8.15 *	6.67 *

The Table 4A shows the percentage of total apoptotic cells. A Wilcoxon test was used for statistical analysis. *p* < 0.05 was considered statistically significant compared to the untreated control (NT) (Table 4B). A Wilcoxon test was used to evaluate the statistical differences. * *p* ≤ 0.05 compared to NT.

**Table 5 ijms-20-01049-t005:** T-cell viability. The viability of T-cells co-cultured with ITP-MSCs was estimated by a cytofluorimetric assay after 24 h treatment with JWH (2.5 µM) and Dexa (100 nM), alone and in combination.

**T Cell Viability**					
**Sample 1**	**T cells**	**T cells + MSC**	**T cells + MSC** **JWH-133**	**T cells + MSC** **DEXA**	**T cells + MSC** **J + D**
(Number of cells × 10^6^)	6.68	7.01	5.56 *	6.44 *	6.23 *
**Sample 2**	**T cells**	**T cells + MSC**	**T cells + MSC** **JWH-133**	**T cells + MSC** **DEXA**	**T cells + MSC** **J + D**
(Number of cells × 10^6^)	7.20	7.12	5.75 *	6.37 *	6.11 *

The table shows the results, as cell number per 10^6^. A Wilcoxon test was used for statistical analysis. *p* < 0.05 was considered statistically significant. * vs. T-cells + MSC.

**Table 6 ijms-20-01049-t006:** TNF-α release quantification. The release of the pro-inflammatory TNF-α by T-cells alone and co-cultured with ITP-MSCs was investigated by ELISA assay after 24 h treatment with JWH (2.5 µM) and Dexa (100 nM), alone and in combination. LPS (500 ng/mL) was administrated to each sample before treatments.

**TNF-α**							
**Sample 1**	**T cells**	**T cells + MSC**	**T cells + LPS**	**T cells + MSC LPS**	**T cells + MSC** **L + JWH-133**	**T cells + MSC** **L + DEXA**	**T cells + MSC** **L + J + D**
pg/mL	44.61	31.53 *	43.11 ^	45.26 ^	39.38 °	28.89 °	27.48 °
**Sample 2**	**T cells**	**T cells + MSC**	**T cells + LPS**	**T cells + MSC** **LPS**	**T cells + MSC** **L + JWH-133**	**T cells + MSC** **L + DEXA**	**T cells + MSC** **L + J + D**
pg/mL	45.82	32.68 *	42.91 ^	45.03 ^	40.67 °	29.41 °	30.05 °

The table shows the concentrations (pg/mL) of TNF-α. A Wilcoxon test was used for statistical analysis. *p* < 0.05 was considered statistically significant. * vs. T-cells; ^ vs. T-cells + MSC; ° vs. T-cells + MSC LPS.

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
