# Peer review of "CB2 Receptor Stimulation and Dexamethasone Restore the Anti-Inflammatory and Immune-Regulatory Properties of Mesenchymal Stromal Cells of Children with Immune Thrombocytopenia"

_ijms, 2019, doi:10.3390/ijms20051049_

Reviewer 1 Report

The authors in the manuscript entitled "CB2 receptor stimulation and Dexamethasone restore the anti-inflammatory and immune-regulatory properties of Mesenchymal Stromal Cells of children with Immune Thrombocytopenia" intended to investigate the possibility that MSCs from ITP patients are responsive targets for the anti-inflammatory effects of dexamethasone. Combining Dexa with JWH-133, an agonist of CB2 receptor that has been demonstrated by the same authors to mediate the immunosuppressive activities of hMSCs, better restored the anti-inflammatory and immune-regulatory properties of ITP-MSCs. They claimed that it's possible that combining with JWH-133 reduces the Dexa dose and thus the side effects but maintains its therapeutic benefits on the impaired immunosuppressive capacity of ITP-MSCs.

1. Since the reduced expression of CB2 in ITP-MSCs (Fig 2) is negatively correlated with secretions of proinflammatory cytokines (IL-17, IL-6, INF-gamma) and positively with anti-inflmmatory cytokine (IL-4) secretion, it most likely means that CB2 alone (a minimal production of endocannabinoids) is capable of inhibiting/promoting pro-/anti-inflammatory cytokines. The authors needs to clarify whether treatments of LPS, JWH-133, and Dexa affect CB2 expression or simply its activity. Was CB2 also responsible for the restoration effects of Dexa? Or they were CB2-independent activities?

2. Silencing CB2 in healthy MSCs or overexpressing CB2 in ITP-MSCs should be done to corroborate whether CB2 is truly required for the anti-inflammatory and immune-regulatory properties of ITP-MSCs.

3. In Fig. 3, the authors failed to perform important controls, i.e., ITP+J, ITP+D, and ITP+J+D, as ITP NT already secreted significantly more/less amount of pro-/anti-inflammatory cytokines. Since ITP-MSCs were isolated from ITP patients who already had prominent inflammation, what was the point of treating ITP-MSCs with LPS? J and/or D should be competent to reduce/increase pro-/anti-inflammatory cytokines without needing stimulation by LPS.

4. In Fig. 5, it was not clear even in Materials and Methods how T cell viabilities were measured when T cells were mixed with MSCs in the co-culture system.

5. In Fig. 6, how did authors know whether TNF-a was released by T cells or MSCs when they were mixed in the co-culture system? One control was missing, i.e., MSC+LPS, to determine whether MSCs alone were able to release TNF-a in response to LPS.

Author Response

Response to Reviewer 1 Comments

We thank the reviewer for his/her comments and for the opportunity to improve our manuscript.

Point 1: Since the reduced expression of CB2 in ITP-MSCs (Fig 2) is negatively correlated with secretions of pro-inflammatory cytokines (IL-17, IL-6, INF-gamma) and positively with anti-inflammatory cytokine (IL-4) secretion, it most likely means that CB2 alone (a minimal production of endocannabinoids) is capable of inhibiting/promoting pro-/anti-inflammatory cytokines. The authors need to clarify whether treatments of LPS, JWH-133, and Dexa affect CB2 expression or simply its activity. Was CB2 also responsible for the restoration effects of Dexa? Or they were CB2-independent activities?

Response 1: In ITP-MSCs, JWH-133 treatment is able to induce an over expression of the CB2 receptor (data not shown). We analysed CB2 expression only after treatment with JWH-133 so it is likely that the effect seen is due to both CB2 increased expression and activity. We did not evaluate whether LPS and Dexa affect CB2 expression. We hypothesize independent activities for JWH-133 and Dexa, and as mentioned in the manuscript (lines 267-270), certainly further investigations are needed to clarify if there is any interaction between the two drugs.

Point 2: Silencing CB2 in healthy MSCs or overexpressing CB2 in ITP-MSCs should be done to corroborate whether CB2 is truly required for the anti-inflammatory and immune-regulatory properties of ITP-MSCs.

Response 2: Surely the silencing of CB2 would be useful to strengthen our results, however it would require some months while we aim to make our findings available to the scientific community as soon as possible and we have a very restricted time frame for revisions (only 10 days). Nevertheless, in a previous study we investigated the involvement of CB2 in healthy MSCs viability and cytokine release, observing that the selective CB2 stimulation with JWH-133, was able to mediate their inflammatory and immunosuppressive properties. Moreover, to confirm the activation of the CB2 receptor and its role, we also stimulated CB2 in presence of the selective CB2 antagonist, AM630, which strongly counteracted all the effects induced by JWH-133 (Rossi F. et al., 2013 Plos one). To strengthen our conclusions and address your concern, we induced an overexpression of CB2 stimulating it with JWH-133 in ITP-MSCs, demonstrating that the receptor is truly required for their anti-inflammatory and immune-regulatory properties considering that in these cells, prior to treatments, we observed a reduced expression of CB2. Moreover, in the revised version of our manuscript, we show in supplementary figures 1, 2A and supplementary table 1 the effects of JWH-133 in presence of the CB2R inverse agonist, AM630, on cytokine release, MSCs apoptosis and T cell viability observing a strongly reduction of its effects.  This result adds a further evidence about the role of the CB2 receptor as a mediator of the inflammatory and immunosuppressive properties of MSCs. Even if the receptor blockage is not comparable to silencing, certainly, it strengthens our hypothesis.

Point 3: In Fig. 3, the authors failed to perform important controls, i.e., ITP+J, ITP+D, and ITP+J+D, as ITP NT already secreted significantly more/less amount of pro-/anti-inflammatory cytokines. Since ITP-MSCs were isolated from ITP patients who already had prominent inflammation, what was the point of treating ITP-MSCs with LPS? J and/or D should be competent to reduce/increase pro-/anti-inflammatory cytokines without needing stimulation by LPS.

Response 3: As suggested, in the revised version of our manuscript, we show in supplementary figure 1 the effects of JWH-133 and Dexa, alone and in combination, on cytokine release without LPS stimulation, observing as expected that both JWH-133 and Dexa are able to restore the cytokine balance. We have stimulated ITP-MSCs with LPS to standardize the inflammatory state of enrolled patients and mend for possible differences due to a different individual autoimmune response as explained in the section “Discussion” (lines 227-229).

Point 4. In Fig. 5, it was not clear even in Materials and Methods How T cell viabilities were measured when T cells were mixed with MSCs in the co-culture system.

Response 4: Considering that MSCs grow in adhesion and T cells in suspension, after the treatments, T-cells were easily isolated simply aspiring them from the co-culture. To make it clearer, in the revised version of our manuscript, we better explain how T cells are isolated from co-culture prior to the Count and Viability Assay (Materials and Methods 4.8 Count and Viability Assay lines 390-391) and we reviewed the title of Figure 5 because it was misleading.

Point 5. In Fig. 6, how did authors know whether TNF-a was released by T cells or MSCs when they were mixed in the co-culture system? One control was missing, i.e., MSC+LPS, to determine whether MSCs alone were able to release TNF-a in response to LPS.

Response 5: In a previous study we have already shown that healthy MSCs are able to release TNF-α and that this release increased in response to LPS (Rossi F. et al., 2013 Plos one). In ITP-MSCs the release of TNF-α is not detectable compared to that of the T cells, also after LPS stimulation (Supplementary figure 2B). This observation leads us to hypothesize that the TNF quantified, is released by T cells.

We acknowledge that this explanation might be not enough to support our hypothesis, so in our revised manuscript (section discussion lines 258-261) we explain that the result observed is only hypothesis-generating and we review the title of figure 6 because it was confounder. 

Reviewer 2 Report

The authors, Rossi and colleagues presented an interesting study revealing that both CB2 receptor expression (i.e. mRNA levels) and density (protein levels) are reduced in ITP-MSCs.

Albeit there is a lot of work put in this paper and the findings have the potential to be of high importance, there are substantial flaws with the design that practically fail to support the main message.

The “n” was 2 control and 2 ITP patients. I wonder what statistical program allows calculating a t-test from n=2, and note that figures can be only produced from data as low as n=3. N=2 prompts the use of data tables instead of bar graphs.

CB2R antibodies are notoriously non-specific. Currently, this reviewer does not know any selective CB2R antibody that gives little or no staining in CB2R KO tissue or no specific band in tissue lisates. This does not mean that no CB2R antibody can produce selective signal under special circumstances and I also do not say that the bands presented in this MS are not for the CB2R, but it would be great if the authors could provide such evidence, if nothing else, by citing the selective use of their antibodies in other studies where they were tested in KO or in CBR-nonexpressing systems.

See also:

Zhang et al. (2018) CB2 receptor antibody signal specificity: correlations with the use of partial CB2-knockout mice and anti-rat CB2 receptor antibodies. Acta Pharmacol Sin. doi: 10.1038/s41401-018-0037-3. Please note that the term expression is reserved only for mRNA – since it tells about gene expression, while proteins have densities based on the WB.

The authors swapped the concentrations of the two compounds… 2.5 microM of JWH133 is 5-times higher than the maximum that I would use, as this compound has an EC50 ~600 nM for the CB1R and ~20 nM for the CB2R. At 2.5 microM, JWH133 is not selective to the CB2R. If the authors did not find effect for JWH133 at lower concentrations then it is because it had its effect via the CB1R already. JWH133's effects should have been tested in the presence of the CB2R antagonist, AM630 (1 uM). In comparison, dexa could have been used at higher concentration, taken that in humans, it is used in a very high dose in certain conditions, and 100 nM is way below the cellular concentration of dexa when it is given in bolus.

Lines 143-144: 143 The same effect was observed when the two drugs were used in combination.

The authors show here occlusion between the effect of the two drugs. These data can be interpreted in 3 ways. 1) these Western blots were not optimize for greater signal intensity and the signal is already saturated, that is, this protocol won’t be able to distinguish among high protein densities, or 2), the system is already pushed to the maximum in terms of Bcl2 production, and there cannot be additivity between JWH133 and dexa, or, it is indeed the correct result, and dexa’s effect is converging onto the CB2 receptor, that is, dexa is releasing endocannabinoids, acting at the CB2 receptor. See e.g. Hill, M.N.; McEwen, B.S. Endocannabinoids: The silent partner of glucocorticoids in the synapse. Proc. Natl. Acad. Sci. USA, 2009, 106, 4579-4580.  and Bowles NP et al., 2015, A peripheral endocannabinoid mechanism contributes to glucocorticoid-mediated metabolic syndrome. Proc Natl Acad Sci U S A. 2015 Jan 6;112(1):285-90. doi: 10.1073/pnas.1421420112.

There was similar occlusion on cell viability, however, there was additivity in the drug effect on TNFalpha levels.

Minor:

Following IUPHAR guidelines, the correct term is cannabinoid CB2 receptor, with the “2” in lower case.

The manuscript is peppered with grammar mistakes, typos and cumbersome text to be corrected, preferably by a native speaker or MS Word corrector turned on. E.g. line 37: phonotype; line 40: lymphocytes proliferation; line 48: macrophages, that; line 50: cytokines balance; line 69: variation of CB2 expression (instead: reduced CB2 expression) etc.

Author Response

Response to Reviewer 2 Comments

We thank the reviewer for his/her comments and for the opportunity to improve our manuscript.

Point 1. The “n” was 2 control and 2 ITP patients. I wonder what statistical program allows calculating a t-test from n=2, and note that figures can be only produced from data as low as n=3. N=2 prompts the use of data tables instead of bar graphs.

Response 1: We did not find any fundamental objection to the use of a regular t-test with extremely small sample sizes. However, to address the reviewer objection we performed a Wilcoxon test, which is a non-parametric test that is more statistically sound to analyze very small datasets. The Test confirmed the significance previously obtained. We revised the manuscript accordingly, mentioning the different statistical test used in Materials and Methods and in the figure legends.

Point 2. CB2R antibodies are notoriously non-specific. Currently, this reviewer does not know any selective CB2R antibody that gives little or no staining in CB2R KO tissue or no specific band in tissue lysates. This does not mean that no CB2R antibody can produce selective signal under special circumstances and I also do not say that the bands presented in this MS are not for the CB2R, but it would be great if the authors could provide such evidence, if nothing else, by citing the selective use of their antibodies in other studies where they were tested in KO or in CBR-non expressing systems.

See also:

Zhang et al. (2018) CB2 receptor antibody signal specificity: correlations with the use of partial CB2-knockout mice and anti-rat CB2 receptor antibodies. Acta Pharmacol Sin. doi: 10.1038/s41401-018-0037-3. Please note that the term expression is reserved only for mRNA – since it tells about gene expression, while proteins have densities based on the WB.

 Response 2: In several studies we evaluated the density of the CB2 protein by western blot and immunofluorescence (Rossi F et al; 2009 Bone; 2010 Bone; 2013 Pharmacological Research; Bellini G et al; 2017 Current Cancer Drug). In the years, we tested almost all CB2 commercially available antibodies. The CB2R antibody used in this study and also in other previous studies is the anti-CB2 from abcam, catalogue number: 3561 experimentally validated in our laboratory and also tested by Zhang et al. in CB2-KO mouse. To be more specific and to facilitate the reproduction of the experiment described, we added in the revised Manuscript, the catalogue number of the CB2R antibody used and we mentioned the study suggested by the Reviewer. Moreover, we changed the term expression in densities when we talk about proteins. 

Point 3. The authors swapped the concentrations of the two compounds… 2.5 microM of JWH133 is 5-times higher than the maximum that I would use, as this compound has an EC50 ~600 nM for the CB1R and ~20 nM for the CB2R. At 2.5 microM, JWH133 is not selective to the CB2R. If the authors did not find effect for JWH133 at lower concentrations, then it is because it had its effect via the CB1R already. JWH133's effects should have been tested in the presence of the CB2R antagonist, AM630 (1 uM). In comparison, dexa could have been used at higher concentration, taken that in humans, it is used in a very high dose in certain conditions, and 100 nM is way below the cellular concentration of dexa when it is given in bolus.

 Response 3: The concentrations of the compounds used have not been swapped and were determined through pilot dose–response experiments. We used the concentrations producing the strongest effect without showing toxicity on the cells. However, to address the reviewer suggestion, we tested the effects of JWH-133 in presence of the CB2R inverse agonist, AM630 (1 uM) on cytokine release, MSCs apoptosis and T cell viability demonstrating a strongly reduction of its effects. (Supplementary figures 1A-B, 2A and supplementary table 1). Dexa concentration was also chosen referring to present literature. As demonstrated in several studies, different concentrations of Dexa exert diverse effects on MSC proliferation, apoptosis and cytokine release. A low dose of Dexa favors MSC expansion in vitro and protects against apoptosis while high dose could damage immunosuppression of MSCs (Wang H. et al; 2012, Cytotherapy; Wang H. et al; 2018, Cell Transplant). However, the clinical practice demonstrated us that it is not possible to make any correlation between in vivo and in vitro doses. Moreover, each in vitro study can require different doses according to the cell type used and cell culture conditions. Our experience proved us that the best way to choose a compound concentration to use, is to perform a time-dependent dose-response curve prior to each study, and for each drug to use.

Point 4. Lines 143-144: 143 The same effect was observed when the two drugs were used in combination.

The authors show here occlusion between the effect of the two drugs. These data can be interpreted in 3 ways. 1) these Western blots were not optimize for greater signal intensity and the signal is already saturated, that is, this protocol won’t be able to distinguish among high protein densities, or 2), the system is already pushed to the maximum in terms of Bcl2 production, and there cannot be additivity between JWH133 and dexa, or, it is indeed the correct result, and dexa’s effect is converging onto the CB2 receptor, that is, dexa is releasing endocannabinoids, acting at the CB2 receptor. See e.g. Hill, M.N.; McEwen, B.S. Endocannabinoids: The silent partner of glucocorticoids in the synapse. Proc. Natl. Acad. Sci. USA, 2009, 106, 4579-4580.  and Bowles NP et al., 2015, A peripheral endocannabinoid mechanism contributes to glucocorticoid-mediated metabolic syndrome. Proc Natl Acad Sci U S A. 2015 Jan 6;112(1):285-90. doi: 10.1073/pnas.1421420112.

There was similar occlusion on cell viability, however, there was additivity in the drug effect on TNF alpha levels.

Response 4: It is not clear to us what exactly the reviewer refers to. The cited line 144 referred to Figure 5, in which we showed the effect of ITP MSCs on T cells viability pre- and after treatments. We are not sure about what the reviewer refers to with the term “occlusion”, however in our opinion, data in Figure 5 clearly show that both drugs restore the MSCs capability to inhibit T cells proliferation. This restoring effect is evident also after co-administration. We did not investigate the mechanism underlying these effects, but in our discussion we explained that in literature the role of GC in reducing T cell growth is already known, so our data “only” confirmed this activity. The aspect we wanted to highlight with our study, is that we observed a more marked effect when Dexa is used together with a selective agonist at CB2 receptor. This result generates the hypothesis of a synergism (not occlusion) between the two drugs, even though further investigations are requested (line 235-239). As regards the Western Blotting on Bcl-2, the Reviewer mention 3 possible interpretations of data, but he/she does not mention the third one. We do not see saturation; hence it is possible to distinguish the different bands density using the “Image studio Digits ver. 5.0” software”. On the other hand, regarding the specific target of Dexa, we do not find in literature any evidence about Dexa property to release endocannabinoids in the peripheral nervous system.

Minor:

Point 1. Following IUPHAR guidelines, the correct term is cannabinoid CB2 receptor, with the “2” in lower case.

Response 1: We changed it in the revised version of our manuscript.

Point 2. The manuscript is peppered with grammar mistakes, typos and cumbersome text to be corrected, preferably by a native speaker or MS Word corrector turned on. E.g. line 37: phonotype; line 40: lymphocytes proliferation; line 48: macrophages, that; line 50: cytokines balance; line 69: variation of CB2 expression (instead: reduced CB2 expression) etc.

Response 2: We corrected the grammar mistakes as suggested.

Round  2

Reviewer 1 Report

1. Compared with CTR NT (high level of CB2 R), MSC ITP NT (low level of CB2 R) was enough for significantly inhibiting/promoting pro-/anti-inflammatory cytokines, indicating that CB2 protein itself has the functionality in the absence of CB2 ligand (Figure 3). In that sense, what needs to be done is only to overexpress CB2 protein level, but not to use activity agonist JWH-133. Although JWH-133 was able to upregulate CB2 protein, it is a chemical compound and may have CB2-independent off target effects on cytokine secretion. The same reason is applied to my previous suggestion in which I suggested "Silencing CB2 in healthy MSCs or overexpressing CB2 in ITP-MSCs should be done to corroborate whether CB2 is truly required for the anti-inflammatory and immune-regulatory properties of ITP-MSCs ". Therefore, the title should be changed to “ CB2 receptor overexpression---------“, because I don’t see any role of CB2 activity stimulated with the CB2 ligand. Thus, the suggested experiments are required to be done to support the revised title. Using CB2 antagonist AM630 (supplementary figures 1, 2A and supplementary table 1) may well have off-target effects and won't be conclusive enough. I suggest that the authors request the editor an extended time frame and also directly transfect healthy MSC cells with CB2 siRNA to significantly shorten the limited time frame.

2. I don't really agree that treatment of LPS is necessary for the standardization purpose as LPS is not present in ITP patients anyway. Besides, judging from Fig. 1~3, the differences between MSC CTR and ITP were significant enough. Fig 3 should be totally replaced by Fig S1 with additional results for IL-17 and INF-gamma. LPS should not be used to treat T cells as well in Fig. 6.

3. According to Fig S1, neither additive nor synergistic effects for combining J and D were observed. What's the point of doing such combination experiments and how could patients benefit from them?

Author Response

Response to Reviewer 1 Comments

We thank the reviewer for his/her comments and for the opportunity to improve our manuscript.

Point 1: Compared with CTR NT (high level of CB2 R), MSC ITP NT (low level of CB2 R) was enough for significantly inhibiting/promoting pro-/anti-inflammatory cytokines, indicating that CB2 protein itself has the functionality in the absence of CB2 ligand (Figure 3). In that sense, what needs to be done is only to overexpress CB2 protein level, but not to use activity agonist JWH-133. Although JWH-133 was able to upregulate CB2 protein, it is a chemical compound and may have CB2-independent off target effects on cytokine secretion. The same reason is applied to my previous suggestion in which I suggested "Silencing CB2 in healthy MSCs or overexpressing CB2 in ITP-MSCs should be done to corroborate whether CB2 is truly required for the anti-inflammatory and immune-regulatory properties of ITP-MSCs ". Therefore, the title should be changed to “CB2 receptor overexpression---------“, because I don’t see any role of CB2 activity stimulated with the CB2 ligand. Thus, the suggested experiments are required to be done to support the revised title. Using CB2 antagonist AM630 (supplementary figures 1, 2A and supplementary table 1) may well have off-target effects and won't be conclusive enough. I suggest that the authors request the editor an extended time frame and also directly transfect healthy MSC cells with CB2 siRNA to significantly shorten the limited time frame.

Response 1: We discussed this point with the Editor and concluded not to proceed with other experiments. 

Point 2: I don't really agree that treatment of LPS is necessary for the standardization purpose as LPS is not present in ITP patients anyway. Besides, judging from Fig. 1~3, the differences between MSC CTR and ITP were significant enough. Fig 3 should be totally replaced by Fig S1 with additional results for IL-17 and INF-gamma. LPS should not be used to treat T cells as well in Fig. 6.

Response 2: As suggested, in the revised version of our manuscript we replaced Fig. 3 with Fig. S1. Considering the significant differences observed between CTR and ITP MSC in cytokine release, we removed any reference to LPS in the manuscript according to the reviewer’s comment.

However, in order to analyze the TNF-α release, we believe that LPS treatment is instead necessary to mimic inflammatory condition in T cells, since these cells were isolated from healthy subjects.

Point 3. According to Fig S1, neither additive nor synergistic effects for combining J and D were observed. What’s the point of doing such combination experiments and how could patients benefits from them?

Response 3: We hypothesize independent activities for JWH-133 and Dexa, and as mentioned in the manuscript, certainly further investigations are needed to clarify if there is any interaction between the two drugs. Our data suggest the possibility to use Dexa in combination with CB2 stimulation in ITP, reducing its dose, therefore its side effects maintaining its therapeutic benefits.

Reviewer 2 Report

The authors made some improvements to the paper with the use of the CB2R inverse agonist AM630 which is great. Nevertheless, the major problem stays, i.e. n=2 is not to be shown in bar graph. I studied Statistics at the university, and one of the things we learnt was that if you have two separate samples you show them separately (or put them in a table), and only from n=3 you can make graph out of it.

I understand that there was no chance to improve the "n" but I also have to pretend to be a scientifically sound reviewer and point out what I would never do in my papers.

Occlusion means that one compound causes an effect, an another compound causes the exact same effect, and the two combined still causes the exact same effect. This means that either the WB was already saturated not allowing detecting an even greater change, or the effect of the compounds converge on one single route, i.e. the glucocorticoid-cannabinoid axis (as indicated by the references given in the first reviewer opinion). This second case is called an occlusion.

Finally, I do not understand why the authors changed grammatically correct sentences to incorrect ones such as In literature it is widely reported instead of "Several studies suggested that" etc. Also, please note that there is no such thing as reverse agonist.

Author Response

Response to Reviewer 2 Comments

The authors made some improvements to the paper with the use of the CB2R inverse agonist AM630 which is great. Nevertheless, the major problem stays, i.e. n=2 is not to be shown in bar graph. I studied Statistics at the university, and one of the things we learnt was that if you have two separate samples you show them separately (or put them in a table), and only from n=3 you can make graph out of it

I understand that there was no chance to improve the "n" but I also have to pretend to be a scientifically sound reviewer and point out what I would never do in my papers

Occlusion means that one compound causes an effect, an another compound causes the exact same effect, and the two combined still causes the exact same effect. This means that either the WB was already saturated not allowing detecting an even greater change, or the effect of the compounds converge on one single route, i.e. the glucocorticoid-cannabinoid axis (as indicated by the references given in the first reviewer opinion). This second case is called an occlusion.

We thank the reviewer for his/her comments and for the opportunity to improve our manuscript.

As suggested, in the revised version of our manuscript we changed the images from bar graph to tables and showed the two separate samples individually.

Finally, I do not understand why the authors changed grammatically correct sentences to incorrect ones such as in literature it is widely reported instead of "Several studies suggested that" etc. Also, please note that there is no such thing as reverse agonist.

The sentence indicated was modified upon another reviewer’s request.
